# High-resolution imaging of the osteogenic and angiogenic interface at the site of murine cranial bone defect repair via multiphoton microscopy

Kevin Schilling[1,2†], Yuankun Zhai[1†], Zhuang Zhou[1], Bin Zhou[3], Edward Brown[2], Xinping Zhang[1]*

[1]Center for Musculoskeletal Research, University of Rochester, School of Medicine and Dentistry, Rochester, United States; [2]Department of Biomedical Engineering, University of Rochester, Rochester, United States; [3]Shanghai Institutes for Biological Sciences, Shanghai, China

**Abstract** The spatiotemporal blood vessel formation and specification at the osteogenic and angiogenic interface of murine cranial bone defect repair were examined utilizing a high-resolution multiphoton-based imaging platform in conjunction with advanced optical techniques that allow interrogation of the oxygen microenvironment and cellular energy metabolism in living animals. Our study demonstrates the dynamic changes of vessel types, that is, arterial, venous, and capillary vessel networks at the superior and dura periosteum of cranial bone defect, suggesting a differential coupling of the vessel type with osteoblast expansion and bone tissue deposition/remodeling during repair. Employing transgenic reporter mouse models that label distinct types of vessels at the site of repair, we further show that oxygen distributions in capillary vessels at the healing site are heterogeneous as well as time- and location-dependent. The endothelial cells coupling to osteoblasts prefer glycolysis and are less sensitive to microenvironmental oxygen changes than osteoblasts. In comparison, osteoblasts utilize relatively more OxPhos and potentially consume more oxygen at the site of repair. Taken together, our study highlights the dynamics and functional significance of blood vessel types at the site of defect repair, opening up opportunities for further delineating the oxygen and metabolic microenvironment at the interface of bone tissue regeneration.

*For correspondence:
xinping_zhang@urmc.rochester.edu

[†]These authors contributed equally to this work

Competing interest: The authors declare that no competing interests exist.

## Editor's evaluation

In this manuscript, the authors present exciting findings on cranial bone defect repair using cutting-edge multiphoton imaging to study the role of different vessel subtypes and related oxygen and metabolic microenvironments. This allows one to understand the pathophysiological progressions of bone diseases and regeneration. It will also provide critical information to optimize the therapeutic bone healing and regeneration approach for different clinical situations.

## Introduction

Bone defect repair is a well-orchestrated process that requires coordinated and concerted activities from both osteogenic and angiogenic cells (*Hankenson et al., 2011*; *Ai-Aql et al., 2008*; *Grosso et al., 2017*; *Hu and Olsen, 2017*; *Schott et al., 2021*). While the interdependent role of osteogenesis and angiogenesis is well documented, the cellular and molecular events that take place at the osteogenic and angiogenic interface of bone defect repair remain superficially understood. A

functional blood vessel network consists of arteries, veins, and a capillary interface that connect arterial and venous vessels for proper vascular perfusion. Following injury, blood vessels are severed, and angiogenesis commences in response to a low state of oxygen, that is, hypoxia. Localized hypoxia stimulates endothelial cells (ECs) to undergo a series of changes including sprouting of the tip cells, proliferation of the stalk cells, and expansion of the continuous lumen, ultimately forming a functional capillary network to restore nutrients and oxygen supply to the site of the defect. This process occurs concurrently with migration, proliferation, and differentiation of osteoblasts and their progenitor cells at the site of repair. How functional blood vessel networks are re-built and remodeled in response to injury to support the proliferation and differentiation of bone forming cells at an early stage of healing and a late stage of bone tissue remodeling remain poorly understood.

Skeletal vasculature has been shown to be uniquely patterned to adapt to the need and functionality of bone tissues at different anatomic sites (*Watson and Adams, 2018*; *Sivaraj and Adams, 2016*). Recent studies have further shown that bone forming progenitors and osteoblasts are associated with a specialized subset of ECs that expresses high levels of endothelial markers CD31 and endomucin. Type H ($CD31^{high}Emcn^{high}$) and type E ($CD31^{high}Emcn^{low}$) vessels, as opposed to type L ($CD31^{low}Emcn^{high}$) vessels, are shown coupling to $OSX^+$ osteoblasts at the long bone metaphysis and $Gli1^+$ bone progenitor cells at the cranial bone (*Kusumbe et al., 2014*; *Ramasamy et al., 2014*; *Rindone et al., 2021*). Both type E and type H vessels are derived from $Apln^{CreER+}$ vessels that directly connect to arterioles, exhibiting high flow rate and are further regulated by molecular pathways that control EC specification during development, that is, hypoxia inducible factor, BMP, and Notch pathways (*Sivaraj and Adams, 2016*; *Langen et al., 2017*; *Xu et al., 2018*). In view of this recent development, understanding bone-dependent vessel formation and patterning and delineating specific bone progenitor cell interaction with the vascular microenvironment are critically important for developing successful strategies aimed at modulating osteogenesis and angiogenesis at the bone healing interface for enhanced repair and regeneration.

Mapping functional vessel networks in bone has been a persistent challenge due to heavy mineralization of the bone tissue, compounded by the technical difficulty in imaging functional vasculature deep inside tissues. We have previously demonstrated an imaging platform for high-resolution analyses of cranial bone tissue vascularization via multiphoton laser scanning microscopy (MPLSM) (*Zhai et al., 2021*; *Huang et al., 2015*; *Schilling et al., 2019*; *Zhang, 2018*). Compared with confocal microscopy which utilizes a pinhole to eliminate out-of-focus fluorescence, MPLSM fluorescence excitation is intrinsically confined to a very small focal volume, permitting detection of a great deal of scattered fluorescence light. Since multiphoton excitation utilizes a longer wavelength laser to excite fluorescent molecules, penetration of the laser beam into the tissues is also considerably deeper, with further benefits of reduced photobleaching and photodamage (*Hoover and Squier, 2013*; *Niesner and Hauser, 2011*). In addition to imaging fluorescence excitation, MPLSM can also be used for imaging collagen matrix and bone tissue through second harmonic generation (SHG), making it a powerful imaging platform for visualization and analysis of vascularization in bone tissue (*Pendleton et al., 2020*; *Vielreicher et al., 2017*; *Chaudhary et al., 2016*; *Ranjit et al., 2015*).

A series of two-photon or multiphoton-based non-invasive optical techniques have been developed that permit interrogation of physiological, functional, and molecular properties of blood vessels at a high spatiotemporal resolution. Combining phosphorescence quenching with two-photon laser scanning microscopy, an optical technique known as two-photon phosphorescence lifetime microscopy (2P-PLIM) has been established. Equipped with two photon excitable oxygen probes, namely platinum porphyrin coumarin-343 (PtP-C343), this technique enables high spatial resolution measurements of partial oxygen tension ($pO_2$) in living tissue (*Finikova et al., 2008b*; *Lecoq et al., 2011*; *Sakadžić et al., 2010*; *Spencer et al., 2014*). Another non-invasive technique developed to obtain information on cell metabolism and to differentiate different metabolic states of cells and tissues is the fluorescence lifetime-dependent measurements of free and bound NAD(P)H via two-photon microscopy (2P-FLIM) (*Chance, 2004*; *Mayevsky and Chance, 2007*; *Winkler and Hirrlinger, 2015*; *Ma et al., 2016*; *Stringari et al., 2011*; *Stringari et al., 2012b*; *Stringari et al., 2015*; *Wright et al., 2012*). This method exploits the intrinsic autofluorescence lifetimes of free and bound NAD(P)H, which reflect the metabolic state of single cells within the native microenvironment of the living tissue (*Stringari et al., 2011*; *Stringari et al., 2012b*; *Stringari et al., 2015*; *Stringari et al., 2012a*; *Yaseen et al., 2009*; *Kolenc and Quinn, 2019*; *Datta et al., 2020*). By analyzing the lifetime contributions

of enzyme-bound and non-bound fluorescent species, one can infer cellular energy metabolism of a specific cell population in vitro (*Sharick et al., 2018*; *Chacko and Eliceiri, 2019*), ex vivo (*Skala et al., 2007*), as well as in vivo (*Castro et al., 2018*; *Schilling et al., 2022*). These techniques have provided unprecedented opportunities for elucidating the molecular and cellular events, as well as the microenvironmental factors at the biological interfaces of bone tissue repair and regeneration.

The overall goal of our current study was to utilize a series of advanced optical imaging techniques in conjunction with transgenic animals that label subtypes of ECs to define dynamic changes of vessel specification and function at the osteogenic and angiogenic interface during cranial bone defect repair. Our study was motivated by the clinical need to develop effective therapeutics, as well as the importance of addressing a critical knowledge gap in our understanding of the functional vascular bed in bone regeneration. Through high-resolution imaging of the osteogenic and angiogenic interface at the defect site via multiphoton-based microscopy, we highlight the functional significance of the blood vessel types in bone repair, offering new insights into bone tissue vascularization and bone healing microenvironment at the osteogenic and angiogenic interface of bone defect repair.

## Results
### CD31 and EMCN define different types of vessels that displayed distinct dynamics in response to injury at the regenerative interface

Using high-resolution MPLSM, which allows for simultaneous imaging of bone via SHG, donor osteoblasts via Col1a1-GFP, and blood vessels via immunostaining of CD31 and EMCN in the repair tissues, we examined the spatiotemporal specification of the vessel network during the healing of a 1-mm cranial full-thickness defect created in Col1a1-GFP mice over a period of 21 days. As shown (*Figures 1 and 2*), CD31 and EMCN marked two distinct types of vessels at the site of cranial defect: CD31$^+$EMCN$^+$ (merging red and green channel to show as yellowish green or yellow) and CD31$^+$EMCN$^-$ (red only with absence of EMCN staining). A relatively stronger staining of CD31 and EMCN in double positive vessels was observed at the site of active bone repair as compared to the regions of un-injured bone. To gain a better understanding of the vascularization process during repair, all samples harvested at the indicated time points were examined via MPLSM from both superior (*Figure 1*) and dura side of the cranial bone (*Figure 2*). As shown, vessel organization and response at the superior and dura side of cranial bone were distinct. From the superior side, where the top 100–150 µm of the repair tissue was examined over a time course of 21 days (*Figure 1*, panels A, B, C, D, and E), we found few vessels between days 1 and 5 on the surface of bone following injury. A large number of CD31$^+$EMCN$^+$ vessels sprouted out of the defect at day 5. The vessels were further spread out and formed a connected network over the superior surface of bone, suggesting that the majority of the vessel networks at the superior side of cranial bone were originated from dura side periosteum. Compared to CD31$^+$EMCN$^+$ vessels (*Figure 1*, panel E1–7), CD31$^+$EMCN$^-$ vessels (*Figure 1*, panel D1–7) followed a similar trend with significantly more vessels shown at days 14 and 21 around newly formed bone at the leading edge of the bone defect. Col1a1-GFP$^+$ osteoblasts (pseudocolored as cyan) appeared at the leading edge of the bone defect (*Figure 1*, panel C1–7) and some along the channels of the existing bone at day 10 (*Figure 1F1, G1*), in accordance with the appearance of some large diameter CD31$^+$EMCN$^+$ vessels in bone tissue. Many of these CD31$^+$EMCN$^+$ vessels were observed to be lined by Col1a1-GFP$^+$ osteoblasts and intertwined with CD31$^+$EMCN$^-$ vessels at days 10, 14, and 21 in newly formed bone undergoing active remodeling (*Figure 1*, panels F and G, arrows).

More detailed analyses were performed on the dura side of the cranial defect where relatively smaller damage occurred to its vasculature. As shown, sporadic osteoblast colonies were found at the dura side periosteum in non-injured bone (*Figure 2D1*). Distinct from the superior periosteum where fewer large vessels were found on the surface, major CD31$^+$EMCN$^+$ and CD31$^+$EMCN$^-$ vessels can be readily identified at the dura periosteum via CD31 and EMCN immunofluorescence staining, suggesting that dura periosteum was better vascularized (*Figure 2*, panel A1-F1). Healing was initiated following injury at day 1 with visibly larger osteoblast colonies seen around the injury site (*Figure 2*, panels A2–F2). The expansion of Col1a1-GFP$^+$ cells (*Figure 2B* 1-7 and *Figure 2D* 1-7, pseudocolored as cyan) peaked at days 5 and 10 post injury, followed by progressively enhanced SHG (*Figure 2A* 1-7, gray to white), indicating progressive bone formation on bone surface. Examination of

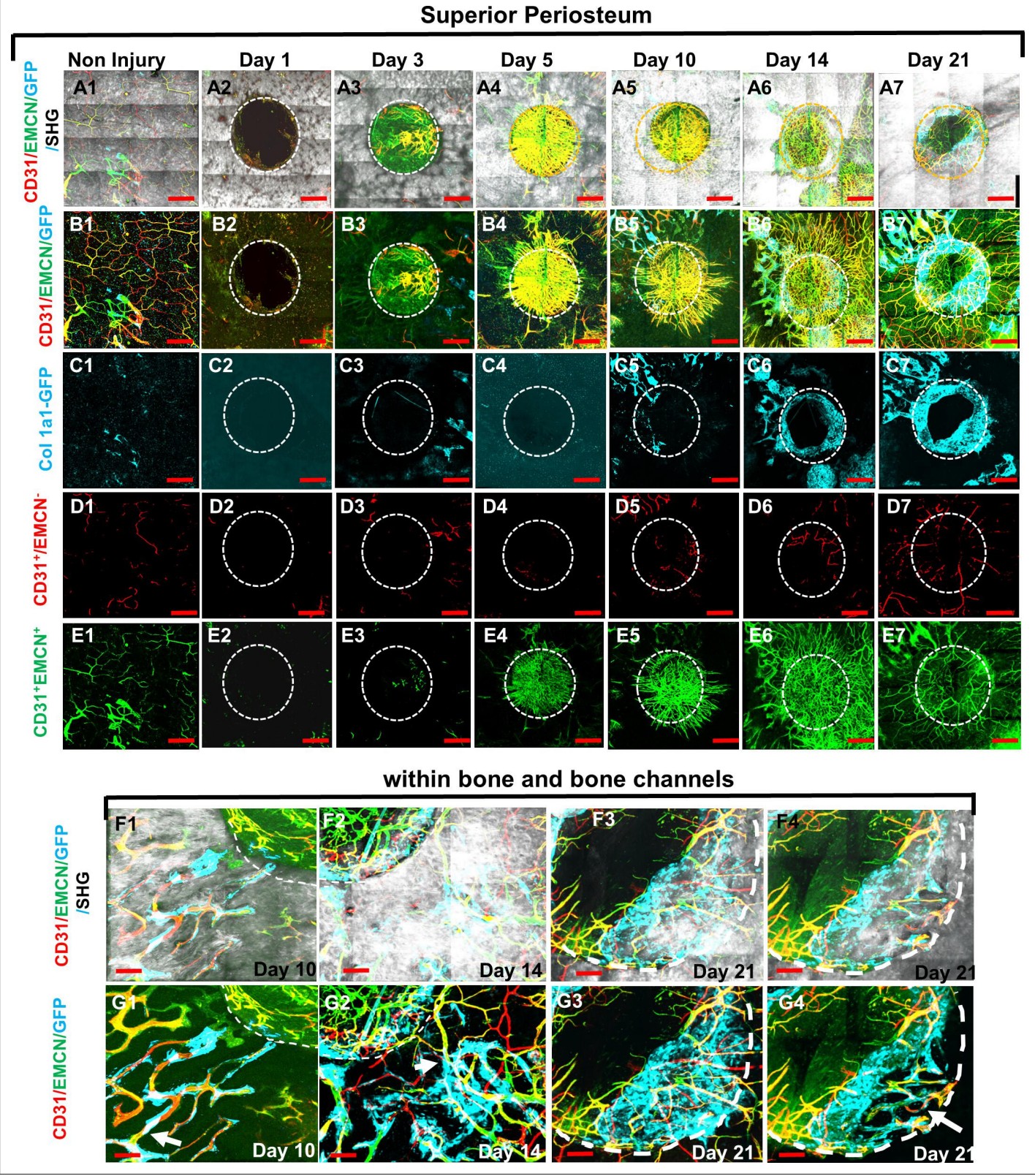

**Figure 1.** Multiphoton laser scanning microscopy (MPLSM) imaging at the superior periosteum of the cranial bone defect in Col1a1-GFP transgenic mice. Cranial defects were imaged via MPLSM from superior side of cranial bone before and after injury at days 1, 3, 5, 10, 14, and 21. Two types of vessels CD31+EMCN+ (yellow or yellowish green), CD31+EMCN− (red), and Col1a1-GFP+ osteoblasts (cyan) were reconstructed with (A1–7) or without bone (gray, B1–7) to illustrate the spatiotemporal changes of vessels during cranial repair at the superior side of the defect. Vessels sprouted from

*Figure 1 continued on next page*

*Figure 1 continued*

the dura side periosteum were abundant in the defect region at day 5. The superior periosteum had no sign of bone formation until day 10. GFP⁺ osteoblasts (cyan) were found primarily at the leading edge of the defect and within bone channels at days 10, 14, and 21 (C1–7). CD31⁺EMCN⁻ (red, D1–7), CD31⁺EMCN⁺ (green, E1–7) were separately reconstructed to show temporal changes of the two types of vessels. Zoom-in images of bone defect at the leading edge and within bone channels are shown at days 14 and 21 with (F1–4) and without bone (G1–4). Arrows show two types of vessels within bone channels, many tightly associated with Col1a1-GFP⁺ cells at day 14 and day 21. Scale bar (panels A–E) = 500 µm, Scale bar (panels F and G) = 100 µm. CD 31 staining is pseudocolored as red, EMCN as green, merged color as yellow or yellowish green, Col1a1-GFP as cyan. Bone via second harmonic generation (SHG) as gray.

The online version of this article includes the following source data for figure 1:

**Source data 1.** Original images for *Figure 1*.

---

the spatiotemporal changes of the two types of vessels showed that CD31⁺EMCN⁺ vessels (*Figure 2*, panel C1–7 indicated as yellow, and panel F1–7 indicated as green) initiated rapid sprouting and expansion as early as day 3 within the defect and in the regions closely associated with Col1a1-GFP⁺ cells. The majority of these vessels showed strong staining of both CD31 and EMCN. The expansion of these CD31⁺EMCN⁺ vessels was supported by significantly increased diameter of the main venule at the peak of angiogenesis. Interestingly, the rapid expansion of CD31⁺EMCN⁺ vessels was transient and quickly reverted back to normal density as the expansion of Col1a1-GFP cells subsided. Compared with CD31⁺EMCN⁺ vessels, branching and extension of CD31⁺EMCN⁻ vessels (*Figure 2*, panel C1–7 shown as red, and panel E1–7 as red) were significantly delayed and the induction only observed at day 10 when new bone tissue started building up and further underwent remodeling. Quantification of the volume fractions of SHG, GFP⁺ cells, CD31⁺EMCN⁺, and length fraction of CD31⁺EMCN⁻ vessels showed distinct changes in the two types of vessels associated with osteoblasts and bone at the initiation, expansion, and bone accruement/remodeling phases of healing over a time course of 21 days (*Figure 2G–L*, n=3). Consistently, the expansion of CD31⁺EMCN⁺ vessels peaked at day 5, coinciding with the expansion of GFP⁺ cells. In comparison, the extension and branching of CD31⁺EMCN⁻ vessels coincided with enhanced SHG, indicating that increased CD31⁺EMCN⁻ vessels were associated with bone tissue deposition and remodeling.

## Apln^CreER labels CD31⁺EMCN⁺ capillary vessels whereas Bmx^CreER labels CD31⁺EMCN⁻ vessels at the regeneration interface

To further establish the identity of the microvessels at the site of the defect, tamoxifen (TM) inducible transgenic mouse lines targeting sprouting ECs (*Apln^CreER*) (*Langen et al., 2017*; *He et al., 2016*; *Liu et al., 2015*) and arterial ECs (*Bmx^CreER*) (*Ehling et al., 2013*) were used to trace the vessel identity. To visualize *Apln^CreER*- and *Bmx^CreER*-labeled vessels, we generated *Apln^CreER*; *Ai6* (Zsgreen) and *Bmx^CreER*; *Ai14* (tdTomato) mice. Following two consecutive injections of TM prior to surgery, vessels were analyzed at day 14 post injury following counterstaining with CD31 and EMCN. As shown (*Figure 3*, panel A1–8), *Apln^CreER*-labeled vessels were completely overlapping with CD31⁺EMCN⁺ vessels, and excluded from CD31⁺EMCN⁻ vessels within the defect (*Figure 3A5*) and in bone channels (*Figure 3A8*). In contrast, in a similar experiment, *Bmx^CreER*; *Ai14* reporter mice labeled nearly exclusively arterioles and capillary vessels descending from arterioles in the healing bone tissue (*Figure 3*, panel B1–4). These vessels, shown as red, were mutually exclusive from EMCN⁺ vessels (shown as green) at the defect (*Figure 3B1–3*) or in bone (*Figure 3B4*). These experiments demonstrated that CD31⁺EMCN⁺ vessels were derived from *Apln^CreER+* sprouting ECs whereas CD31⁺EMCN⁻ vessels extended from *Bmx^CreER+* arterial ECs.

## Spatiotemporal analyses of oxygen microenvironment via 2P-PLIM

To obtain a deeper understanding of the impact of vessel types on the oxygen microenvironment at the osteogenic and angiogenic interface during defect repair, we performed intravital and longitudinal measurements of $pO_2$ in a cranial defect window chamber model (*Huang et al., 2015*; *Schilling et al., 2019*) using respective reporter mice that label the two different types of vessels during healing. In order to simultaneously image vessels coupling with osteoblasts, we crossed *Apln^CreER* reporter mice with Col1a1-GFP mice to generate a triple transgenic mouse line, *Apln^CreER*;Ai14;Col1a1-GFP. Through genetic labeling of *Apln^CreER+* vessels with RFP (tdTomato) and osteoblasts with GFP, we were able to examine the $pO_2$ within *Apln^CreER+* vessels and interstitial space near clusters of osteoblasts.

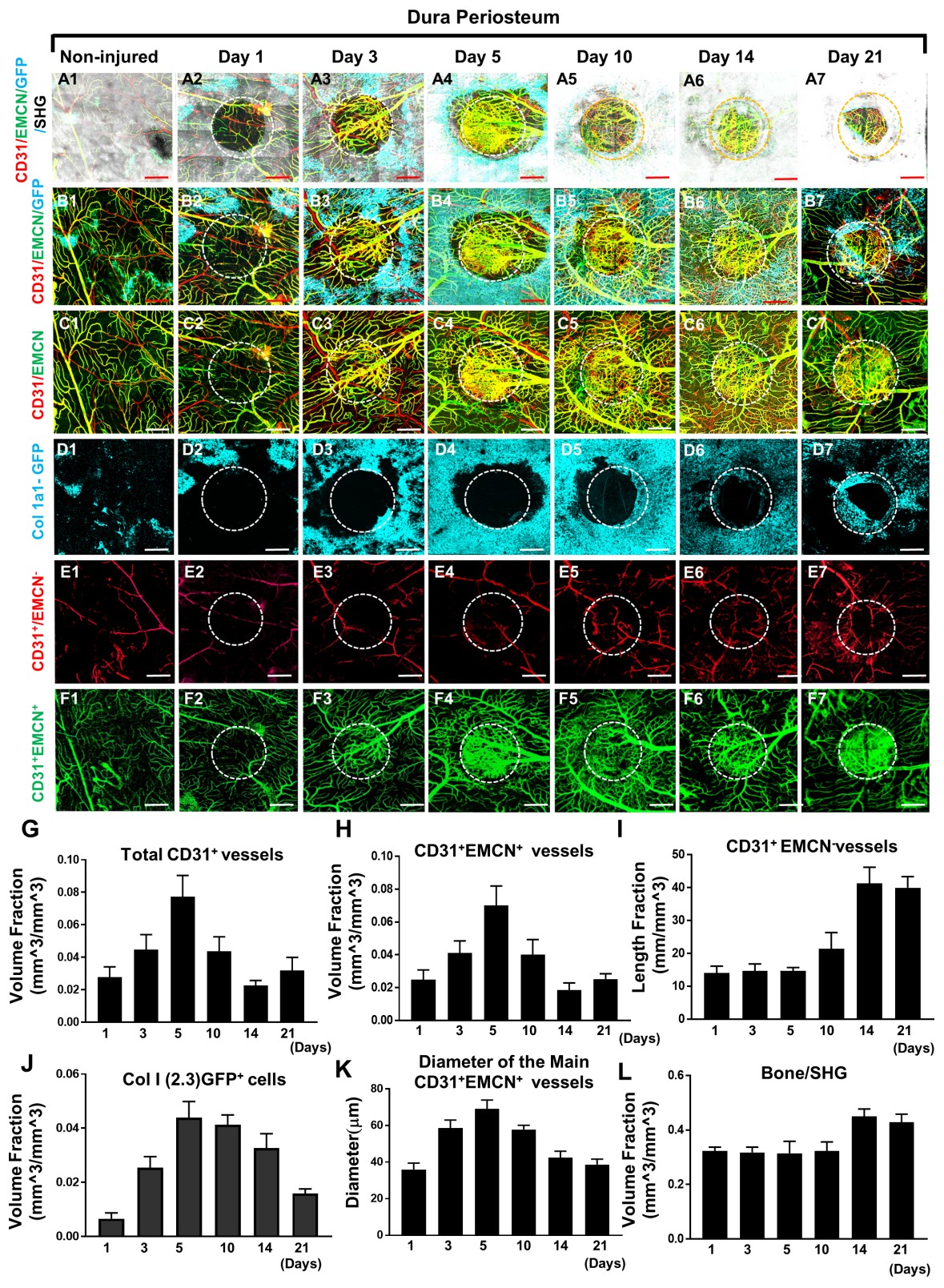

**Figure 2.** Multiphoton laser scanning microscopy (MPLSM) imaging at the dura periosteum of the cranial bone defect in Col1a1-GFP transgenic mice. Cranial defects were imaged from the dura side of the cranial bone via MPLSM before and after injury at days 1, 3, 5, 10, 14, and 21. The reconstructed images illustrate bone (gray), Col1a1-GFP osteoblasts (cyan), CD31$^+$EMCN$^+$ vessels (yellow or greenish yellow), and CD31$^+$EMCN$^-$ vessels (red) at the dura side of the defect (A1–7). To show spatiotemporal relationship of bone, osteoblasts and two types of vessels, the images were reconstructed to

*Figure 2 continued on next page*

*Figure 2 continued*

separately illustrate GFP$^+$ osteoblasts with CD31$^+$EMCN$^+$ and CD31$^+$EMCN$^-$ vessels (B1–7), two types of vessels together (C1–7), GFP$^+$ osteoblasts only (D1–7), and CD31$^+$EMCN$^-$ only (E1–7) or CD31$^+$EMCN$^+$ (green) only vessels (F1–7). Quantitative analyses show volume fraction of the total CD31$^+$ (**G**), CD31$^+$EMCN$^+$ vessels (**H**), Length fraction of CD31$^+$EMCN$^-$ vessels (**I**), the associated changes of volume fraction of Col1a1-GFP osteoblasts (**J**), the mean diameter of the main CD31$^+$EMCN$^+$ vessels (**K**) and the volume fraction of bone as indicated by second harmonic generation (SHG) (**L**). n=3. Scale bar = 500 μm. CD 31 staining is pseudocolored as red, EMCN as green, merged color as yellow or yellowish green, Col1a1-GFP as cyan. SHG/bone as gray.

The online version of this article includes the following source data for figure 2:

**Source data 1.** Original images for *Figure 2*.

Longitudinal imaging was conducted at days 10, 17, 24, and 31 post-surgery (*Figure 4*, panels A and B). Similar to the healing dynamics shown in *Figure 1* and as we have described previously (*Huang et al., 2015*), vessels were observed as early as day 10 in the defect while clusters of GFP$^+$ cells were observed at the leading edge. Significant expansion of GFP$^+$ osteoblasts occurred at the leading edge of the defect at around days 10–17 post-surgery followed by progressive healing as indicated by SHG.

Multiple pO$_2$ measurements were made via point scans of PtP-C343 phosphorescence in various regions of the defects over a period of 31 days. Representative images obtained at a defect region at day 10 (*Figure 4C–G*) illustrated the pO$_2$ measurements in *Apln$^{CreER+}$* vessels within the defect (*Figure 4C*) and those closely associated with the clusters of osteoblasts at the leading edge of the bone defect (*Figure 4D* boxed region and *Figure 4E–G*). Given the fact that oxygen diffusion distance in tissue is ~150 μm (*Jain, 1999*; *McGuire and Secomb, 2003*; *Koike et al., 2004*), pO$_2$ measurements in vessels and interstitium less than or greater than 150 μm distance from GFP$^+$ osteoblast clusters were compared. As shown, significantly lower pO$_2$ levels were found at days 10, 17, and 31 post surgery in vessels and interstitium less than 150 μm from clusters of osteoblasts (*Figure 4H1 and K*, p<0.05). Of note is that pO$_2$ measurements were recorded at the lowest levels at day 17 (mean = 23.69 ± 10.39 mmHg) (p<0.05), indicating a hypoxic environent, presumably due to an increased metabolic need associated with the expansion of osteoblasts in the newly forming bone. Interestingly, the pO$_2$ in all areas gradually increased as more vessel network formed and became mature over the 31-day period.

Similar imaging analyses were conducted at days 10, 17, 24, and 31 post-surgery in the *Bmx$^{CreER}$;Ai14* mice which label arterial ECs (*Figure 5*). As shown (*Figure 5*, A1–A4), the defect underwent progressive healing over a period of 31 days, during which only arterial vessels were labeled, a pattern that was distinctly different from *Apln$^{CreER}$*-labeled vasculature. To characterize the oxygen microenvironment associated with these vessels, regions were selected to report pO$_2$ longitudinally as previously described. Intra-vascular pO$_2$ measurements made within the defect at day 10 were significantly higher (mean = 49.66 ± 17.97 mmHg) than those in the defect interstitium (mean = 35.81 ± 10.8 mmHg, *Figure 5B*, p<0.05). Similar to the *Apln$^{CreERT2}$;Ai14;Cola1-GFP* mice, pO$_2$ was lowest at day 17 in vessels in new bone and defect interstitium. The *Bmx$^{CreER+}$* vessels had higher pO$_2$ than the nearby interstitial space at days 10, 17, and 31 post-surgery (*Figure 5B*, *, p<0.05). Measurements of pO$_2$ significantly increased within all regions with the highest levels recorded from the vessels in the defect at ~74.68 ± 7.64 mmHg at day 31. Bmx$^+$ vessels in new bone and the defect interstitium were ~61.58 and~55.35 mmHg, respectively.

Additional spatial analyses were performed based on pO$_2$ measurements and the distance from the measured points to the initial edge of the bone defect. All data from vessels and interstitial measurements were combined (*Figure 5*, C1-4). Pearson correlation analyses showed no correlation between pO$_2$ and the distance from the edge of the bone defect at days 10 and 17. However, a significant correlation was found at days 24 and 31, indicating that at the later stages of healing, there was a slightly lower environmental pO$_2$ along the area of newly formed bone, potentially due to increased oxygen usage at the area (*Figures 5C3 and 4*, day 24: $R$=0.171, p=0.042; day 31: $R$=0.253, p=0.006). It is worth noting that the mean pO$_2$ in *Bmx$^{CreER+}$* vessels was significantly higher than those of *Apln$^{CreER+}$* vessels when measured at comparable time points and locations, with the measurements at days 24 and 31 showing much larger differences (*Figure 5—figure supplement 1*). These data suggest a process of maturation of the blood vessel network over time, with pO$_2$ eventually reaching equilibrium after weeks of healing.

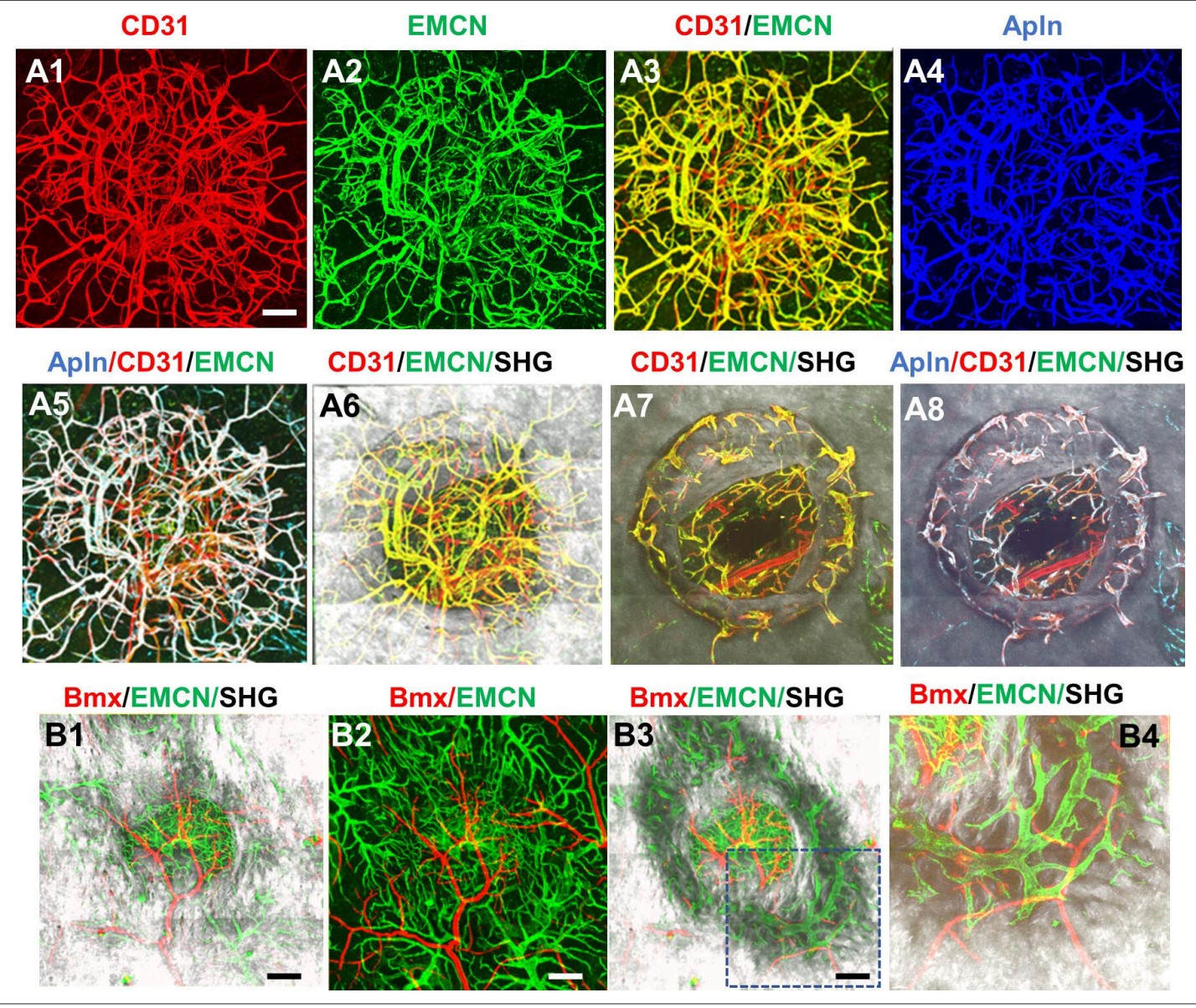

**Figure 3.** *Apln^CreER* and *Bmx^CreER*, respectively, label CD31⁺EMCN⁺ and CD31⁺EMCN⁻ vessels in cranial defect repair. Cranial samples in tamoxifen (TM)-treated *Apln^CreER*; Ai6 mice at day 14 were stained with CD31 and EMCN and reconstructed with various combinations of channels as indicated to reveal the identity of vessel types (A1–8). Noted that *Apln^CreER*-labeled vessels (pseudocolored as blue in A3) are completely overlapping with CD31⁺EMCN⁺ vessels with merged colors shown as white in A5 and 8. *Apln^CreER*-labeled vessels are excluded from CD31⁺EMCN⁻ vessels shown as red. *Bmx^CreER*; *Ai14*-labeled vessels (red) and EMCN⁺ vessels (green) in TM-treated Bmx^CreER; *Ai14* mice at day 14 post-surgery (B1–4). Bmx⁺ vessels (red) and EMCN⁺ vessels (green) are mutually exclusive. Images A1–6 and B1–2 were reconstructed from top to bottom at 0–300 μm in depth whereas images A7–8 and B3–4 were reconstructed from 100 μm below surface to show vessels inside bone tissue. scale bar = 200 μm.

The online version of this article includes the following source data for figure 3:

**Source data 1.** Original files for *Figure 3*.

## Analyses of cellular metabolism in vivo show Apln^CreER+ ECs utilize more glycolysis than osteoblasts

NAD(P)H 2P-FLIM was performed to examine energy metabolism in fluorescently labeled *Apln^CreER+* ECs and osteoblasts during defect repair. It has been well established that a decrease of the bound form of NAD(P)H (lower $\tau_M$) indicates increased glycolysis whereas an increase of the bound form of NAD(P)H (higher $\tau_M$) correlates with increased OxPhos usage (*Kolenc and Quinn, 2019*; *Chacko and Eliceiri, 2019*). Utilizing 2P-FLIM, we previously demonstrate that the osteoblasts at the leading edge

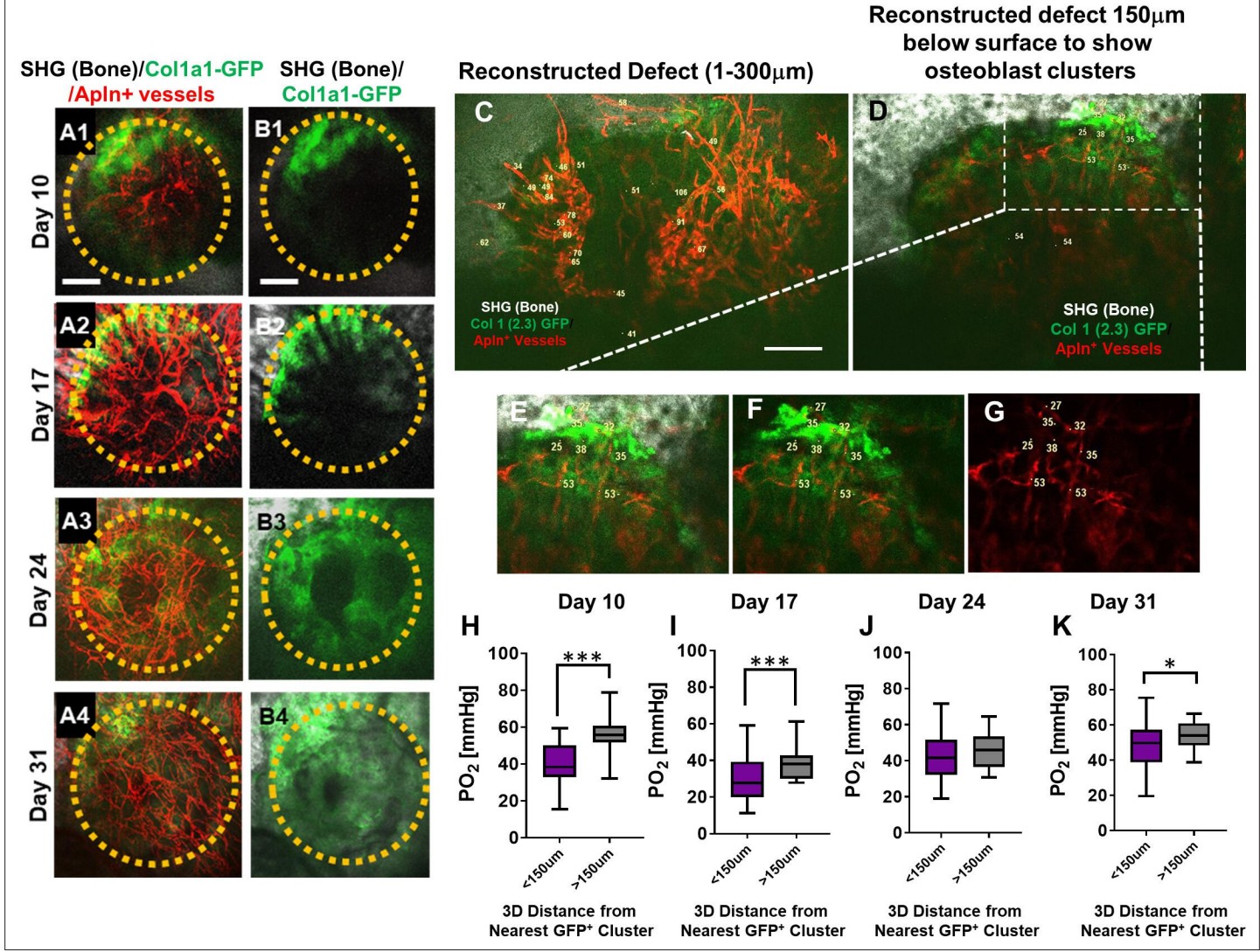

**Figure 4.** Spatiotemporal analyses of partial oxygen pressure (pO₂) using *Apln^CreER*;*Ai14*;Col1a1-GFP mice. Representative maximum intensity projection images of osteoblasts (green), Apln⁺ vessels (red), and bone via second harmonic generation (SHG; gray) of the defect at days 10, 17, 24, and 31 post-surgery (**A1–A4, B1–B4**). The dashed orange line indicates the initial defect (scale bar = 0.25 mm). Representative multiphoton laser scanning microscopy images show vessels and pO₂ measurements at the indicated locations. Images are reconstructed at 0–300 μm (**C**) and 150–300 μm depths (**D–G**) to illustrate vessels and pO₂ measurements with or without osteoblast clusters. Boxed region in D is zoomed-in to show vessels and the GFP⁺ osteoblasts (**E–G**). Box plot to compare measurements of pO₂ less than or greater than 150 μm distance from the closest osteoblast clusters at the indicated time points (**H–K**). (At least 10 measurements in each region at each time point from 4 mice were used for analyses, *, p<0.05, **, p<0.01, ***, p<0.005, ****, p<0.001).

The online version of this article includes the following source data for figure 4:

**Source data 1.** Original data sets and images.

**Source data 2.** Original data sets and images.

of the defect had significantly lower $\tau_M$ than those within the contralateral native bone, suggesting a switch to a more glycolytic state at the time of healing (**Schilling et al., 2022**). To determine whether the pO₂ environment could similarly impact EC energy metabolism during healing, we performed 2P-FLIM in *Apln^CreERT2*;*Ai14*;Col1a1-GFP mice. Since osteoblasts and ECs were labeled with GFP and RFP, respectively, we were able to separately evaluate the cellular metabolism of GFP⁺ osteoblasts and *Apln^CreER⁺* ECs at single-cell resolution via phasor plot analyses to determine NAD(P)H $\tau_M$ in vivo. As shown, the cellular energy metabolism of osteoblasts and ECs was measured using the 2P-FLIM phasor plot (**Figure 6A1–6**). Longitudinal measurement of NAD(P)H $\tau_M$ suggested that *Apln^CreER⁺* ECs had a

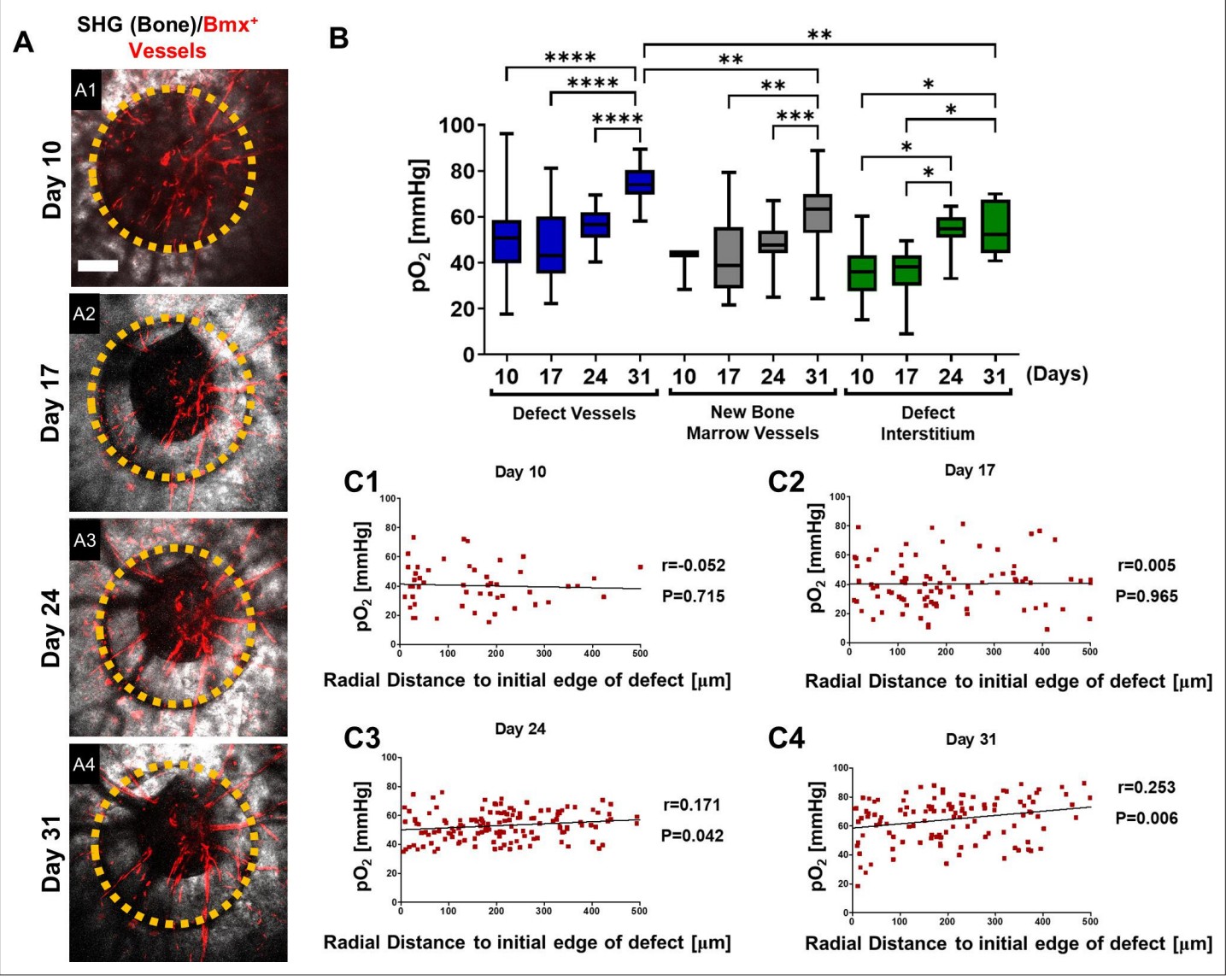

**Figure 5.** Spatiotemporal analyses of partial oxygen pressure (pO₂) using *Bmx^CreER*; *Ai14* mice. Representative maximum intensity projection merged images of Bmx⁺ vessels (red) and bone via second harmonic generation (SHG; gray) of the defect at days 10, 17, 24, and 31 post-surgery (A1–4). The dashed orange line indicates the initial defect (scale bar = 0.25 mm). Box plot of the pO₂ in the indicated time and region at the site of defect repair are shown (**B**). Pearson correlation analyses of the reported pO₂ in vessels and the surrounding interstitial space as a function of the distance to the initial edge of the defect at indicated time points (**C1–4**). (Measurements in each region at each time point from 4 mice are used for analyses, *, p<0.05, **, p<0.01, ***, p<0.005, ****, p<0.001).

The online version of this article includes the following source data and figure supplement(s) for figure 5:

**Source data 1.** Original data set and statistical analyses.

**Figure supplement 1.** Comparison of longitudinal partial oxygen (pO₂) measurements in AplnCreERT2 (green box) and BMX1CreERT2 (red box) by time and locations.

significantly lower NAD(P)H $\tau_M$ than osteoblasts (**Figure 6C**, *P*<0.001), suggesting that compared to osteoblasts *Apln^CreER+* ECs prefer glycolysis over OxPhos. Furthermore, when compared to osteoblasts which shifted the $\tau_M$ to a lower value at day 17 when pO₂ was lower (p<0.001), *Apln^CreER+* ECs were less sensitive to the changes of environmental pO₂ at the same locations (**Figure 6B, C**).

To further determine the cell metabolism of Apln⁺ ECs, we isolated bone marrow from *Apln-CreERT2;Ai14;*Col1a1-GFP mice and cultured it over 24 days (**Figure 6D1–6E**). Measurement of NAD(P)H $\tau_M$ showed that osteoblasts in osteogenic media had an increasing $\tau_M$ longitudinally from 1.55±0.13 ns at day 17 to 1.75±0.18 ns at day 24, whereas cells in regular media had a consistently

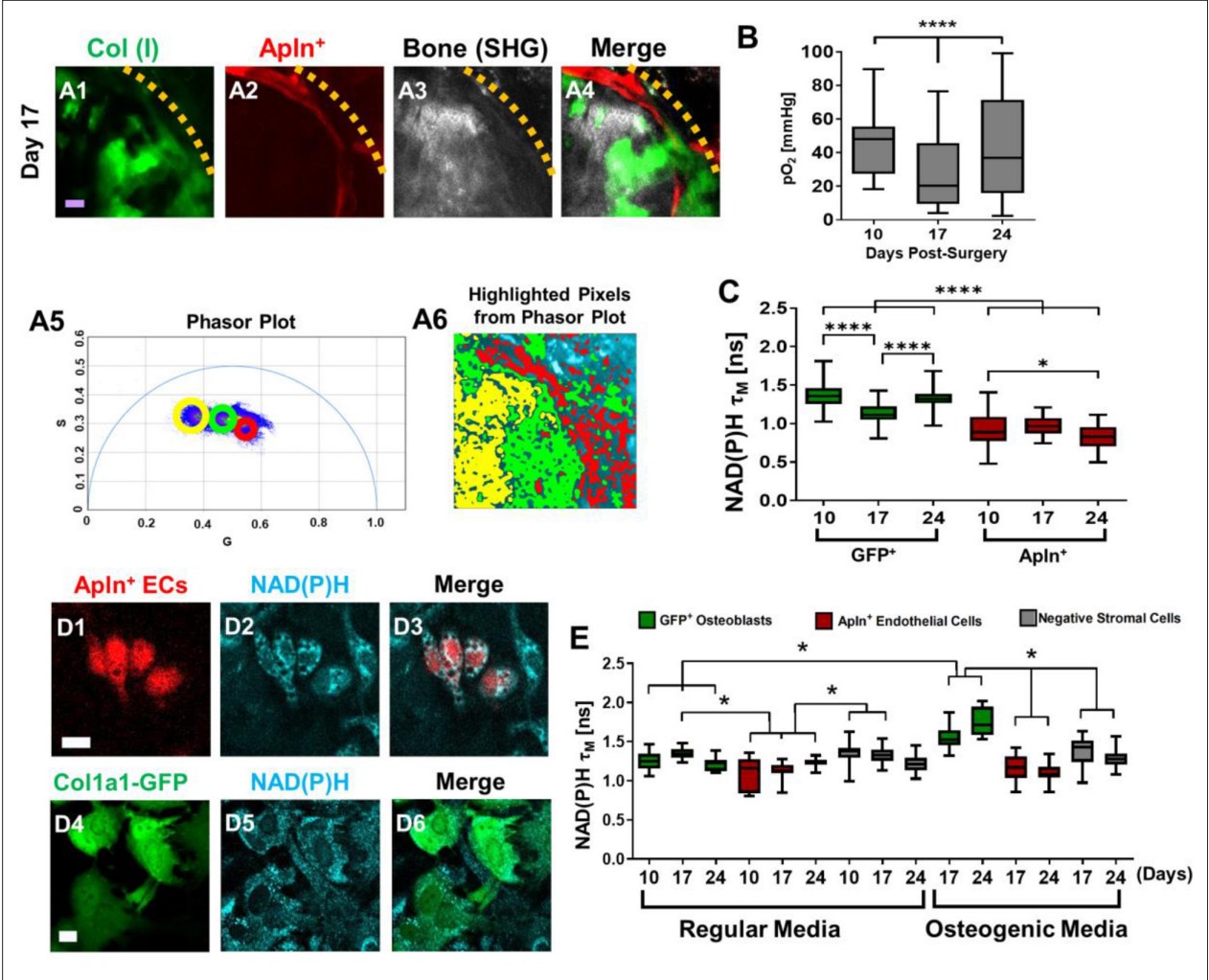

**Figure 6.** Measurements of energy metabolism in endothelial cells (ECs) and osteoblasts via two-photon NAD(P)H fluorescence lifetime imaging microscopy using *Apln^CreER;Ai14;*Col1a1-GFP mice. Representative multiphoton laser scanning microscopy (MPLSM) images of osteoblasts (green), Apln⁺ vessels (red), and bone via second harmonic generation (SHG; gray) in 1-mm cranial defect (**A1-4**). The dashed orange line indicates the initial defect area (scale bar = 20 µm). The phasor plot (**A5**) shows the distribution of the fluorescence lifetime species. The circled lifetime species in A5 correspond to the lifetime species located in *Apln^CreER* ECs (red), osteoblasts (green), and collagen/bone (yellow) as shown in A6. Reported mean partial oxygen pressure (pO₂) from vascular and interstitial measurements within the healing defect (**B**). Box plot of the measurements of NAD(P)H $\tau_M$ of osteoblasts (green) and Apln⁺ ECs (red) at days 10, 17, and 24 post-surgery (**C**). At least 26 cells in each of 3 mice are used for the in vivo NAD(P)H $\tau_M$ analyses. About 22 corresponding pO₂ measurements from each of 3 mice are used for analyses. Representative MPLSM images of bone marrow culture from *Apln^CreERT2;Ai14;*Col1a1-GFP mice are shown (D1–6, scale bar = 5 µm). NAD(P)H from all cells in cyan, RFP from Apln⁺ ECs in red, GFP from osteoblasts in green. Measurements of NAD(P)H $\tau_M$ from GFP⁺ osteoblasts (green box), Apln⁺ ECs (red box) as well as the GFP⁻ stromal cells (gray box) in regular or osteogenic media at days 10, 17, and 24 post-seeding (**E**). At least eight cells in each of three parallel experiments per group are used for the in vitro NAD(P)H $\tau_M$ analyses. *, p<0.05, **, p<0.01, ***, p<0.005, ****, p<0.001.

The online version of this article includes the following source data for figure 6:

**Source data 1.** Original data set and statistical analyses.

similar $\tau_M$ of ~1.3 ns among all time points. Further analyses showed that $\tau_M$ from osteoblasts in osteogenic media at days 17 and 24 was different from osteoblasts in regular media, consistent with our previous findings (*Schilling et al., 2022*). In comparison, *Apln^CreER* ECs had a lower NAD(P)H $\tau_M$ than GFP⁺ osteoblasts and GFP⁻ stromal cells at all time points examined (*Figure 6E*, p>0.05). Additionally, the NAD(P)H $\tau_M$ from *Apln^CreER* ECs was similar in osteogenic and regular media (*Figure 6E*, p>0.05), supporting our finding in vivo which suggests that compared with ECs, osteoblasts used more OxPhos at the site of healing.

## Discussion

To better understand the osteogenesis and angiogenesis coupling during skeletal repair and regeneration, high-resolution MPSLM was employed in conjunction with advanced optical techniques to examine the specification and dynamics of blood vessel types at the osteogenic and angiogenic interface during cranial bone defect repair. Our data show that CD31 and EMCN mark arterial and venous capillary network with CD31$^+$EMCN$^+$ vessels derived from *Apln$^{CreER+}$* sprouting ECs and CD31$^+$EMCN$^-$ vessels from *Bmx$^{CreER+}$* arterial ECs. The two types of vessels demonstrate distinct dynamics in response to injury with osteoblast clusters tightly coupled to CD31$^+$EMCN$^+$ (*Apln$^{CreER+}$*) vessels at the regenerative interface. By probing local pO$_2$ via 2P-PLIM and cellular metabolism via 2P-FLIM, we further show that *Apln$^{CreER+}$* ECs utilize more glycolysis than Col1a1-GFP$^+$ osteoblasts at the site of repair. Energy metabolism of osteoblasts is also more sensitive to pO$_2$ changes than that of *Apln$^{CreER+}$* ECs. Taken together, our current study highlights the functional differences of blood vessel types at the osteogenic and angiogenic interface of bone defect repair, offering imaging tools and new insights for a deeper understanding of bone tissue vascularization and the bone healing microenvironment at the site of regeneration.

CD31 and EMCN have been shown to differentially mark arterial, venous, and capillary vessels in various tissues (*Morgan et al., 1999*; *Samulowitz et al., 2002*; *dela Paz and D'Amore, 2009*; *Zuercher et al., 2012*). Utilizing an intravital MPLSM imaging approach, we have previously shown that CD31$^+$EMCN$^-$ vessels exhibit higher mean pO$_2$ than CD31$^+$EMCN$^+$ vessels at the osteogenic and angiogenic interface during nanofiber-mediated cranial defect repair (*Zhai et al., 2021*). To further understand the dynamics of these two types of vessels during skeletal healing, we performed longitudinal analyses of the two types of vessels during spontaneous cranial defect repair, together with analyses of osteoblasts via Col1a1-GFP and bone matrix deposition via SHG. Our data demonstrate distinct dynamics of the two types of vessels marked by CD31 and EMCN in response to injury: CD31$^+$EMCN$^+$ vessels expand rapidly, coinciding with the expansion of the clusters of Col1a1-GFP$^+$ osteoblasts, while CD31$^+$EMCN$^-$ vessels show a delayed response to injury, with enhanced branching and extension of the vessels observed at a late stage of healing, when formation of vascular channels within bone tissue is evident. The differential dynamics of the angiogenic response reflects the intrinsic differences between venous and arterial ECs in response to bone injury, which could be attributed to the fundamental differences in the biology of the two types of vessels (*dela Paz and D'Amore, 2009*; *Hwa et al., 2017*; *Aitsebaomo et al., 2008*; *Simons and Eichmann, 2015*; *Herbert and Stainier, 2011*). The differential dynamics of angiogenesis from the two types of vessels could also be attributed to the repertoire of angiogenic factors produced by osteoblasts to induce a blood vessel network that is specialized for bone regeneration and homeostasis (*Schipani et al., 2009*; *Stegen and Carmeliet, 2018*; *Stegen et al., 2015*; *Gerstenfeld et al., 2003*). More detailed studies are needed to define vasculature as well as the factors that organize the specialized vessel network in bone tissue.

It is worth noting that the rapid expansion of the CD31$^+$EMCN$^+$ vessels is transient and quickly subsides when osteoblasts become more mature and when significant bone matrix deposition and remodeling follow. The regression of the vessels is necessary for organization of a specialized bone vasculature, specifically the formation of type H vessels in new bone tissue. This finding is consistent with our previous observation which shows similar down-regulation of CD31$^+$EMCN$^+$ vessels following the deposition of bone tissue in a nanofiber-mediated bone regeneration (*Zhai et al., 2021*). This finding is also consistent with more recent studies which link specialized osteoclastic activities with organization and formation of type H vessel formation in bone (*Romeo et al., 2019*; *Kohara et al., 2022*; *Liu et al., 2021*), suggesting that the formation of a specialized vascular network in bone involves concerted activities from osteoblasts, osteoclasts as well as subsets of ECs.

By analyzing the time-dependent changes of bone and vessel formation at the superior and dura periosteum of cranial defect repair, we further show that revascularization at the superior side of cranial bone is largely dependent upon the rapid expansion of the vasculature from the dura periosteum. Similarly, progenitor cells located at the dura periosteum and in the marrow space are the major sources of bone forming cells during healing, as these cells are the first to respond to injury (*Figures 1 and 2*). The differences in the response to injury at the two different sites of periosteum could be attributed to the extent of damage inflicted on vessels and progenitor cells during initial surgery, in which the dura periosteal cells/vessels are better preserved. However, the difference could also be

due to the different density and distribution of vessel types present on the superior and dura periosteum. As a matter of fact, we see very different vessel network in non-injured bone on dura and superior periosteum (*Figure 1B1* vs. *Figure 2B1*) with more organized arterial and venous network formed at the dura side periosteum. More studies are necessary to better characterize the different vessel types and function in associations of osteoblastic progenitors at the dura and superior periosteum.

Apelin, encoded by gene *Apln*, is a highly conserved secreted peptide that binds and acts as an endogenous ligand for a G protein-coupled receptor APJ (*del Toro et al., 2010*). *Apln* is expressed in endothelial tip cells and sprouting ECs in various organs (*Langen et al., 2017*; *Liu et al., 2015*; *Chen et al., 2016*; *Kocijan et al., 2021*) and has been shown to be significantly increased in response to a hypoxic environment (*Ronkainen et al., 2007*). By tracing $Apln^{CreER+}$ vessels during repair, our current study shows that nearly all $CD31^+EMCN^+$ capillary vessels are labeled by $Apln^{CreER}$ including vessels tightly associated with osteoblasts following injury at day 14, excluding $CD31^+EMCN^-$ vessels derived from arterial vessels (*Figure 3A* 1-8). This result is consistent with the previous study which shows that $Apln^{CreER}$-derived vessels give rise to $CD31^{high}EMCN^{high}$ type H and E vessels that are tightly coupled to osteoblasts during bone development (*Langen et al., 2017*), suggesting that $Apln^{CreER}$ is a useful tool to study vessel coupling with osteoblasts. Our study further confirms that $Apln^{CreER+}$ EC robustly labels venules and venous capillaries but not arterial ECs at the site of injury (*He et al., 2016*; *Liu et al., 2015*).

In contrast to Apln, endothelial bone marrow tyrosine kinase *Bmx* is a gene known to be expressed in ECs of arterial vessels (*Rajantie et al., 2001*; *Ekman et al., 1997*). The $Bmx^{CreER}$ mouse model has been shown to largely label arterial vessels during retinal angiogenesis (*Ehling et al., 2013*). Consistent with this, our current study shows that $Bmx^{CreER+}$ vessels are primarily arterial $CD31^+EMCN^-$ vessels directly extended from arterioles at the healing site (*Figures 3B1–4*). $Bmx^{CreER+}$ vessels show higher mean $pO_2$ than $Apln^{CreER+}$ vessels at a similar location, particularly when the vessel network at the defect site becomes more mature and stable at a later stage (*Figure 5—figure supplement 1*). In contrast to $Apln^{CreER+}$ vessels which are abundant and subject to dynamic changes at the site of the defect, $Bmx^{CreER+}$ vessels exhibit a slow response to injury during the course of healing, consistent with what is observed in $CD31^+EMCN^-$ vessels.

Genetic labeling of the $Apln^{CreER+}$ vessels and osteoblastic cells in a triple transgenic reporter mouse allows us to utilize advanced imaging techniques to study the oxygen microenvironment at the osteogenic and angiogenic interface during defect repair. Despite heterogeneous $pO_2$ distribution within blood vessels, our analyses show that $Apln^{CreER+}$ vessels closer to osteoblast clusters have significantly lower $pO_2$ at days 10 and 17 (*Figure 4H, I*), coinciding with osteoblast expansion, indicating enhanced demand for oxygen supply and potentially enhanced consumption of oxygen at the osteogenic and angiogenic interface during early stages of healing. In contrast to $Apln^{CreER+}$ vessels which are often found to be close to osteoblasts, $Bmx^{CreER+}$ vessels at days 10 and 17 do not show changes of $pO_2$ as a function of distance (Figures 5C1–2), suggesting that perhaps the dip of $pO_2$ is vessel type dependent and could be attributed to the close coupling of $Apln^{CreER+}$ vessels with osteoblast clusters at the leading edge of the defect.

Genetic labeling of the $Apln^{CreER+}$ vessels and osteoblastic cells in a transgenic reporter mouse further allows us to examine cellular metabolism at the osteogenic and angiogenic interface utilizing a previously established phasor approach for measurement of NAD(P)H $\tau_M$ via 2P-FLIM (*Schilling et al., 2022*). Our data show that $Apln^{CreER+}$ ECs prefer glycolysis and are less sensitive to microenvironmental $pO_2$ changes at the site of defect repair, consistent with reports that show that glycolysis is the primary energy producing mechanisms in ECs (*Eelen et al., 2018*), and glycolysis is highly favored during sprouting (*De Bock et al., 2013*). By choosing glycolysis for ATP production, ECs allow oxygen diffusion to neighboring cells to support healing. Compared to ECs, osteoblasts have higher NAD(P)H $\tau_M$, therefore utilizing relatively more OxPhos. Osteoblasts further show a significant shift of NAD(P)H $\tau_M$ to glycolysis coinciding with the drop of $pO_2$ at the same location. This is consistent with the data that show low oxygen status is favored by osteoblastic aerobic glycolytic metabolism at the site of bone repair and regeneration (*Lee et al., 2017*). Further analyses of energy metabolism of osteoblasts at different stages as well as other cell types at the defect site could help advance our understanding of the impact of oxygen microenvironment and the hypoxic niche on cellular metabolism at the site of bone healing and homeostasis.

The high complexity of the regenerative microenvironment drives the heterogeneous and non-uniform oxygen distribution and therefore the spatiotemporal differences in the structure and function of blood vessel network during repair and regeneration. In our current study, we find that even within the same type of vessels, there exists a high degree of heterogeneity in $pO_2$ distribution. The heterogeneous $pO_2$ could not be attributed to the heterogeneous vessel diameter distribution at the site of repair since our data showed poor correlation of $pO_2$ value with the vessel diameter at the site of repair (data not shown). The $pO_2$ heterogeneity has critical functional consequences for different cell types in bone and could be important for constitution of various stem cell niches (*Itkin et al., 2016*; *Chen et al., 2020*; *Sivan et al., 2019*). It is also worth noting that while the range of $pO_2$ in CD31+EMCN+ vessels falls within the reported blood $pO_2$ in venous vessels, the mean $pO_2$ in CD31+EMCN− capillary vessels is lower than reported in arterial blood (*Marenzana and Arnett, 2013*), likely due to the active exchange and consumption of oxygen at the repair site.

Blood vessel types and function associated with bone repair and bone tissue engineering remain poorly understood. The limited knowledge hinders further efforts aimed at engineering effective and functional vessel networks for enhanced bone defect repair and regeneration. A deeper understanding of the spatiotemporal control of bone-dependent angiogenesis and bone healing microenvironment is critically important for long-term success of material-based approaches for tissue repair and reconstruction. With the establishment of high-resolution and state-of-the-art optical imaging techniques, our current study has laid the foundation for future studies aimed at better understanding of the osteogenic coupling with EC subtypes at regeneration interface, offering useful tools to study stem/progenitor cell behavior as well as bone healing microenvironment at the site of repair for enhanced skeletal repair and regeneration.

# Materials and methods
## Mouse strains
The Col1a1-GFP transgenic mice, which specifically label mature osteoblasts with GFP, are published previously (*Wang et al., 2006*). The *Apln^CreER* mouse model was generated at Shanghai Institutes for Biological Sciences (*He et al., 2016*). The *Bmx^CreER* mouse model (*Ehling et al., 2013*) was obtained through Ximbio via Material Transfer Agreement. Genomic DNA preparations from mouse ear snips were used to determine the genotypes of each mouse strain via PCR. *Gt(ROSA)26Sortm6(CAG-ZsGreen1)*, also known as *Ai6*, and *Gt(ROSA)26Sortm14(CAG-tdTomato)*, known as *Ai14* mice were purchased from Jackson Laboratory. All in vivo experiments were performed using adult 8–12-week-old animals housed in pathogen-free, temperature and humidity-controlled facilities with a 12-hr day-night cycle in the vivarium at the University of Rochester Medical Center. All cages contained wood shavings, bedding, and a cardboard tube for environmental enrichment. All experimental procedures were reviewed and approved by the University Committee on Animal Resources (Protocol UCAR2009-060). General anesthesia and analgesia procedures were performed based on the mouse formulary provided by the University Committee on Animal Resources at the University of Rochester. The animals' health status was monitored throughout the experiments by experienced veterinarians according to the Guide for the Care and Use of Laboratory Animals outlined by the National Institute of Health. Tamoxifen (Sigma, T5648) was dissolved in corn oil (20 mg/ml) and administered by i.p. injection at day 1 and day 2 prior to surgery (0.1–0.15 mg tamoxifen per gram mouse body weight).

## Multiphoton laser scanning microscopy
An Olympus FVMPE-RS system equipped with two lasers: Spectra-Physics InSightX3 (680 nm-1300nm) and Spectra-Physics MaiTai DeepSee Ti:Sapphire laser (690 nm-1040nm), and 25× water objective (XLPLN25XWMP2, 1.05NA) was used for high-resolution imaging. Images were acquired at 512×512 pixels using resonant scanners with the laser tuned to 780 nm. The fluorescence of GFP, RFP, far-red RFP, and SHG signals was collected with a 517/23 nm, a 605/25 nm, a 665/20 nm, and a 390/20 nm bandpass filters (Semrock), respectively. The 2D slice viewing and 3D reconstruction of the defect were performed using Imaris (Bitplane Inc, Concord, MA) and Amira image analysis software (Visage Imaging, Berlin, Germany). The quantitative and histomorphometric analyses of the neovasculature were performed in a defined region as described below. Vessels within each region were isolated in

Amira Segmentation Editor followed by volumetric analyses and histomorphometric analyses using the Autoskeleton Module combined with Filamental Editor as described below.

## Cranial bone defect and windowed chamber model in mice

Procedures for creating a cranial defect and mounting a glass window for imaging in mice have been previously described (*Huang et al., 2015*; *Zhang, 2018*). Briefly, the mouse under anesthesia had hair removed from surgical site of the skull. A stereotaxic instrument (Stoelting Inc, Wood Dale, IL) was used to stabilize the mouse head for surgery under a dissection microscope. A 1 mm in diameter full thickness defect was created in the parietal bone of mouse calvarium using a same sized Busch inverted cone bur (Armstrong Tool & Supply Company, Livonia, MI). In the experiments using the 1 mm defect, bone wounds were closed, and samples were harvested at the indicated time points for imaging and histologic analyses. To perform intravital imaging during cranial defect healing, a custom-made 0.5 mm thick spacer made of poly (aryl-ether-ether-ketone) was glued onto the skull using cyanoacrylate glue (Loctite; Cat #45404, Düsseldorf, Germany). A glass window was mounted on top of the wound for intravital imaging as previously described (*Huang et al., 2015*).

## Measurement of partial oxygen pressure (pO$_2$) in blood vessels via 2P-PLIM

To examine the oxygen content in various blood vessels, 2P-PLIM was performed in the cranial defect window chamber model, permitting real-time interrogation of pO$_2$ within each vessel at high spatial resolution (*Sakadzić et al., 2010*; *Finikova et al., 2008a*). To measure pO$_2$, 10 μmol of the oxygen sensitive phosphorescence probe PtP-C343 was administered into the circulation via retro-orbital injection. A two-photon microscope with a tunable Mai Tai laser (100 fs, 80 MHz; Spectra-physics, Santa Clara, CA) for excitation and a modified Olympus Fluoview 300 scan unit was used for imaging. Excitation of PtP-C343 was performed at 900 nm. The light transmitted through the dichroic was passed through a 706/167 nm band-pass filter and directed onto a PMT (Hamamatsu R10699, Shizuoka, Japan) and a photon counting system (SR 400, Stanford Research Systems, Sunnyvale, CA) for quantification of PtP-C343 phosphorescence ($\lambda_{max}$ = 680 nm). Raw phosphorescence decay data were fit to a single exponential function after subtraction of the offset, to determine the decay time constant, $\tau$. Using an independently measured calibration curve, $\tau$ was converted into oxygen tension (pO$_2$) as previously described (*Schilling et al., 2019*). Point scans were performed in randomly selected vessels within the defect. A mean from at least two measurements was used to determine the oxygen content within each vessel at each location.

## Immunofluorescent staining of blood vessels and microscopy

Mice were perfused with 4% paraformaldehyde following surgery as previously described. The samples were treated with 3% bovine albumin in PBS containing 0.3% Triton X-100 and then stained with CD31 antibody conjugated with Alexa 647 (1:100 dilution, Biolegend, San Diego, CA) and EMCN antibody conjugated with PE (1:50 dilution, Santa Cruz Biotechnology, Santa Cruz, CA) for 3–4 days at 4°C. The samples were imaged via MPLSM as described above (*Zhai et al., 2021*). Different types of vessels were isolated based on the staining of CD31 and EMCN in the Amira Segmentation Editor, which allows overlaying multiple channels for analyses. Analyses and evaluation of the volume and length of each type of vessels were conducted following segmentation of the type of vessels as previously described (*Huang et al., 2015*; *Zhang, 2018*).

## Two-photon NAD(P)H fluorescence lifetime imaging (2P-FLIM) and cellular energy metabolism index determination

BMSCs were extracted from mice and cultured in a standard six-well plate as previously described (*Zhai et al., 2021*; *Schilling et al., 2022*; *Wang et al., 2018*). At designated time intervals, the dish containing cells were removed from the cell incubator and moved to the multiphoton microscope. MPLSM and 2P-FLIM were conducted in regions where heterogeneous cells were located in vitro and in vivo using an XLPlan N 25 X/1.05 NA Olympus water immersion objective to image GFP$^+$ osteoblasts (850 nm excitation, 536/27 nm emission filter) and RFP-labeled ECs (740 nm excitation, 605/55 nm emission filter). To ensure minimal interference from eGFP or collagen which can be excited at the excitation wavelength for NAD(P)H (740 nm) (*Datta et al., 2020*; *Mamontova et al.,*

*2018*; *Conklin et al., 2009*; *Ranjit et al., 2015*), we chose a bandpass filter of 447/60 nm to prevent eGFP emission bleed-through into the NAD(P)H channel while providing a large bandwidth to collect the NAD(P)H emission. The filtered signal was directed to a PMT (Hamamatsu H7422P-40) connected to a photon counting board (ISS A320 FastFLIM). Time-domain NAD(P)H images were collected at 256×256 pixels with a repetition rate of 100 and a pixel dwell time of 20 μs. The time-domain images were transformed to a phasor plot to graphically depict the lifetime species within an image and separately evaluate signal from cells versus other autofluorescence, such as collagen (*Ranjit et al., 2015*; *Ranjit et al., 2019a*; *Ranjit et al., 2018*; *Ranjit et al., 2019b*). A mean lifetime, $\tau_M$, was calculated for each cell within the image based on the G and S coordinates on the phasor plot as well as the laser modulation frequency, $\omega$ (Eq. 1).

$$\tau_M = \frac{1}{\omega}\frac{S}{G} \tag{1}$$

As fibrillar collagen produces a distinct lifetime that can be readily identified in the phasor plot (*Ranjit et al., 2015*), we were able to highlight pixels in cells versus pixels pertaining to extracellular collagen, allowing $\tau_M$ to be determined from individual osteoblasts, ECs, and other stromal cells using the software provided by ISS.

## Quantitative and histomorphometric analyses of neovascularization at the site of cranial bone defect repair

All samples were scanned from the superior and dura side of the cranial bone for analyses. Only analyses from the dura side periosteum are shown. A detailed method for quantitative analyses of blood vessels at the site of cranial defect repair has been previously described (*Huang et al., 2015*; *Zhang, 2018*). Briefly, CD31$^+$EMCN$^-$ or CD31$^+$EMCN$^+$ vessels along with SHG and Col1a1-GFP cells were reconstructed in a 3D format using a multichannel z-series stack. To analyze spontaneous healing in a 1-mm defect, a three-dimensional cubic region of 2×2 mm (16 tiles of 512×512 z-series stack) up to 200 μm in depth, comprised the defect, and the surrounding area was created as the region of interest (ROI) for quantitative analyses. The vascular network within ROI was subsequently isolated using Amira Segmentation Editor to obtain the final segmented vascular image. Using Amira's AutoSkeleton module, which implements a distance-ordered homotopic thinning operation, the segmented 3D vascular network was further skeletonized to generate a line-based network that was topologically equivalent to the original network. The skeleton was superimposed on the original image to assess the relative accuracy of this method. The final skeletonized vessel network was obtained by manually retracing of the skeletons using Amira's Filamental Editor to remove false segments. Based on the skeletonized network, vessel length fraction (i.e. ratio of vessel length to total volume) was read from the Amira software. Quantitative and histomorphometric analyses of neovasculature as described above were performed simultaneously with volumetric quantification of Col1a1-GFP cells and SHG using Amira segmentation Editor and volumetric analyses protocol. Analyses were performed in a group of three mice, covering the entire defect regions.

### Statistical analyses

Statistical analyses were conducted via GraphPad Prism (GraphPad Prism, San Diego, CA). A normality test was performed on each of the datasets collected. Due to the non-parametric distribution of pO$_2$ measurements in some data sets, a box and whisker plot were used to show the data distribution in *Figures 4–6*. For datasets which included a non-parametric distribution, a non-parametric t-test or non-parametric one-way or two-way ANOVA (Kruskal-Wallis test) with Dunn post-hoc was used to determine the statistical significance. Otherwise, a t-test, an ordinary one-way ANOVA, or an ordinary two-way ANOVA with a Bonferroni post-hoc was utilized on normally distributed data. The Pearson correlation coefficients between groups were analyzed in GraphPad to obtain a p value and the co-efficiency r value. In all analyses, a p-value of <0.05 was considered statistically significant.

## Acknowledgements

This study is supported by grants from the National Institutes of Health R01AR067859, R01DE019902, R21DE026256, R01DE029790, R21AR076056, P30AR069655, the Department of Defense W81XWH-17-1-0011, and the National Science Foundation 2150799. The synthesis of probe PtP-C343 was

supported by the grant U24EB028941 and provided by Dr. Sergei Vinogradov at the University of Pennsylvenia.

## Additional information

### Funding

| Funder | Grant reference number | Author |
|---|---|---|
| National Institute of Arthritis and Musculoskeletal and Skin Diseases | R01AR067859 | Xinping Zhang |
| National Institute of Dental and Craniofacial Research | R01DE019902 | Xinping Zhang |
| National Science Foundation | 2150799 | Edward Brown |
| National Institute of Arthritis and Musculoskeletal and Skin Diseases | R21AR076056 | Xinping Zhang |
| National Institute of Arthritis and Musculoskeletal and Skin Diseases | P30AR069655 | Xinping Zhang |
| National Institute of Dental and Craniofacial Research | R21DE026256 | Xinping Zhang |
| National Institute of Dental and Craniofacial Research | R01DE029790 | Xinping Zhang |
| US Department of Defense | W81XWH17-1-0011 | Edward Brown |

The funders had no role in study design, data collection and interpretation, or the decision to submit the work for publication.

### Author contributions

Kevin Schilling, Conceptualization, Data curation, Formal analysis, Supervision, Funding acquisition, Validation, Investigation, Methodology, Writing – original draft, Project administration, Writing – review and editing; Yuankun Zhai, Data curation, Formal analysis, Methodology, Writing – review and editing; Zhuang Zhou, Data curation, Formal analysis, Methodology; Bin Zhou, Methodology, Writing – review and editing, provide reagents; Edward Brown, Methodology, Writing – review and editing; Xinping Zhang, Conceptualization, Data curation, Supervision, Funding acquisition, Validation, Investigation, Methodology, Writing – original draft, Project administration

### Author ORCIDs

Xinping Zhang http://orcid.org/0000-0002-4429-638X

### Ethics

All in vivo experiments were performed using adult 8 to 12-week-old animals housed in pathogen-free, temperature and humidity-controlled facilities with a 12-hr day-night cycle in the vivarium at the University of Rochester Medical Center. All cages contained wood shavings, bedding and a cardboard tube for environmental enrichment. All experimental procedures were reviewed and approved by the University Committee on Animal Resources (Protocol UCAR2009-060). General anesthesia, and analgesia procedures were performed based on the mouse formulary provided by the University Committee on Animal Resources at the University of Rochester. The animals' health status was monitored throughout the experiments by experienced veterinarians according to the Guide for the Care and Use of Laboratory Animals outlined by the National Institute of Health.

Decision letter and Author response
Decision letter https://doi.org/10.7554/eLife.83146.sa1
Author response https://doi.org/10.7554/eLife.83146.sa2

# Additional files

## Supplementary files
• MDAR checklist

## Data availability
All data source files used to generate each figure are provided as Source data.

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
