## [Editor Report]

In this manuscript, the authors present exciting findings on cranial bone defect repair using cutting-edge multiphoton imaging to study the role of different vessel subtypes and related oxygen and metabolic microenvironments. This allows one to understand the pathophysiological progressions of bone diseases and regeneration. It will also provide critical information to optimize the therapeutic bone healing and regeneration approach for different clinical situations.

---

## [Decision Letter]

**Decision letter after peer review:**

Thank you for submitting your article "High Resolution Imaging of the Osteogenic and Angiogenic Interface at the Site of Cranial Bone Defect Repair via Multiphoton Microscopy" for consideration by *eLife*. Your article has been reviewed by 3 peer reviewers, one of whom is a member of our Board of Reviewing Editors, and the evaluation has been overseen by Mone Zaidi as the Senior Editor. The following individuals involved in review of your submission have agreed to reveal their identity: Liu Hong (Reviewer #2); Anjali P Kusumbe (Reviewer #3).

Essential revisions:

1) Please, address reviewers' comments point-by-point.

2) Please, explain the apparent discrepancies between tissue oxygenation and its metabolism as discussed by reviewer #3.

*Reviewer #1 (Recommendations for the authors):*

It would be helpful to more extensively discuss the translational relevance of the findings.

*Reviewer #2 (Recommendations for the authors):*

This is a well-designed and -performed study to visualize the interface of osteogenesis and angiogenesis in mouse calvarial bone healing in real-time. It is a very well-written manuscript and was built on the authors' previous works to visualize, for the first time, the natural bone healing progression in mouse calvarial defects. Notably, it also includes the analysis of the oxygen and energy metabolism within the defects during bone healing, which provides critical information for future optimization of therapeutic approaches for bone regeneration. This study also provides the foundation for the future to visualize the pathophysiologic progressions of bone diseases, including osteoporosis.

*Reviewer #3 (Recommendations for the authors):*

1: The figures are not properly arranged; for example, Figure 2 shows the details of the timeline, but the labelling of the figure, in this case, is mismatched. Non-injured and Day1 is confusing and not arranged. D31 is written instead of CD31. As per the statement, CD31+EMCN+ and CD31+EMCN- show two distinct vessels in yellowish green/yellow and Red colour, respectively, but it is missing in figures 1 and 2. Figure 3 is not showing CD31+EMCN+ and CD31+EMCN- labelling from A1-A4 as described in the text.

2. Please check the manuscript for typos.

3. As per the study, the author described that vessel organization and healing response after injury at the superior and dura side of the cranial bone is distinct. It is shown with the progression of healing at different time points, but there is a lot of difference in non-injury. In Figure 1 it is shown that vessel organization is poor not only on the site of injury but also in the surroundings but in Figure 2 it seems the opposite.

4. Statistical significance is also missing in Figure 4J.

5. Authors could provide some more evidence/time points to show the different levels of NAD(P)H in response to pO2.

6. Authors measured the pO2 labels after injury or during the healing process and found lower levels on days 10, 17 and 31, but the lowest was observed on day 17 because of the hypoxic environment. As per previous experiments (Figure 2), metabolic activities are higher on day 10 post-injury, and vessel proliferation is associated with osteoblast expansion at the same time. These two experiments contradict each other regarding the lowest level of pO2 on day 17. Why the higher metabolic activities did not affect the pO2 level at day 10, and why is there a similar level of pO2 at day 10 and 24 at <150 µm.

---

## [Author Response]

Reviewer #1 (Recommendations for the authors):It would be helpful to more extensively discuss the translational relevance of the findings.

We have included a paragraph to discuss the future directions as well as the translational indication of the study with great input from Reviewer #2 (see last paragraph in the Discussion on page 17).

Reviewer #2 (Recommendations for the authors):This is a well-designed and -performed study to visualize the interface of osteogenesis and angiogenesis in mouse calvarial bone healing in real-time. It is a very well-written manuscript and was built on the authors' previous works to visualize, for the first time, the natural bone healing progression in mouse calvarial defects. Notably, it also includes the analysis of the oxygen and energy metabolism within the defects during bone healing, which provides critical information for future optimization of therapeutic approaches for bone regeneration. This study also provides the foundation for the future to visualize the pathophysiologic progressions of bone diseases, including osteoporosis.

We greatly appreciate the Reviewer’s comments especially the comments regarding the relevance of this study to translational science.

Reviewer #3 (Recommendations for the authors):1: The figures are not properly arranged; for example, Figure 2 shows the details of the timeline, but the labelling of the figure, in this case, is mismatched. Non-injured and Day1 is confusing and not arranged.

We have aligned the labels so that it matches each figure/image.

D31 is written instead of CD31.

We have made the correction in Figure 2.

As per the statement, CD31+EMCN+ and CD31+EMCN- show two distinct vessels in yellowish green/yellow and Red colour, respectively, but it is missing in figures 1 and 2. Figure 3 is not showing CD31+EMCN+ and CD31+EMCN- labelling from A1-A4 as described in the text.

In both Figures 1and2 top panels, CD31+EMCN+ vessels are shown as yellow or yellowish green after merging of CD31 (pseudo-colored as red) and EMCN (pseudo-colored as green) staining. In the same figures/images, if there is no staining for EMCN, the vessels will be shown as bright red, indicating they are CD31+EMCN-. The composite (merged view) of the two types of vessels, with or without SHG (pseudo-colored as grey) or GFP (pseudo-colored as cyan), are further shown in Figure 1 panel A1-7 and B1-7, Figure 2 panel A1-7, B1-7 and C1-7.

To show single staining with different color schemes will make Figures 1 and 2 un-manageable (i.e. more panels and more complex description of each panel). To simplify, we only show the merged color scheme instead of individual staining in both figures.

Again, we showed the merged color of CD31 and EMCN as yellow in Figure 3 for the sake of easy understanding in our previous version of Figure 3. But with the request from the Reviewer, in this revised version of the manuscript we have re-arranged the images of Figure 3 and added additional images to show single staining of CD31 and EMCN as well as the merged images of multiple channels in the panel A1-8. We have also updated the text in the Result section accordingly.

2. Please check the manuscript for typos.

We have checked and read through the manuscript to correct as many typos as we can identify.

3. As per the study, the author described that vessel organization and healing response after injury at the superior and dura side of the cranial bone is distinct. It is shown with the progression of healing at different time points, but there is a lot of difference in non-injury. In Figure 1 it is shown that vessel organization is poor not only on the site of injury but also in the surroundings but in Figure 2 it seems the opposite.

Your observation is correct on the differences between the superior and dura periosteum in the non-injured bone. These differences are currently being investigated in the lab. The density as well as the type of vessels in superior and dura periosteum are apparently different. We have stated the difference in the result section on page 7 (highlighted) and further discussed this difference in the section of Discussion on page 14 (highlighted).

4. Statistical significance is also missing in Figure 4J.

We found no statistical differences between the two groups at day 24 as shown in Figure 4J.

5. Authors could provide some more evidence/time points to show the different levels of NAD(P)H in response to pO2.

We wish we could do more. However, our graduate student Kevin Schilling has completed his PhD thesis and have moved on to a new position. We are currently updating our microscope which prevents us from performing more time point analyses in a short period of time. Given the additional time and mice needed to complete these experiments, we hope the Reviewer will agree that these experiments are dispensable.

6. Authors measured the pO2 labels after injury or during the healing process and found lower levels on days 10, 17 and 31, but the lowest was observed on day 17 because of the hypoxic environment. As per previous experiments (Figure 2), metabolic activities are higher on day 10 post-injury, and vessel proliferation is associated with osteoblast expansion at the same time. These two experiments contradict each other regarding the lowest level of pO2 on day 17. Why the higher metabolic activities did not affect the pO2 level at day 10, and why is there a similar level of pO2 at day 10 and 24 at <150 µm.

First of all, pO_2_ levels in the defect were very heterogeneous as shown in our data and in our discussion. Secondly, from day 10 to day 24, more cells were recruited into the defect with vessel density in the defect markedly increased over time. At day 10, due to the presence of relatively fewer cells at the defect site, the consumption of oxygen by osteoblasts may not exert a direct impact on the mean oxygen level at the defect site. On day 17, with more osteoblasts being formed plus other oxygen consuming cell types being recruited to the site of injury, we observed a significant drop of the overall oxygen level at the defect site. By day 24, more vessels were being formed and getting matured, resulting in an increased availability of oxygen. Hence, at this time, the differences in pO_2_ were lost. This difference was again noted on day 31 when oxygen equilibrium was reached at the site of repair. Overall, we do not think that these experiments are contradicting to each other. On the contrary, they reflect the complexity of the in vivo microenvironment in the context of healing.

Lastly, we want to mention that our current method to measure pO_2_ is limited to a small focal spot (point scan), precluding us from recording pO_2_ in a large area efficiently. We are in a process of updating our microscope so that we can create a density map of pO_2_ in a larger and better-defined region of interest. We are also collaborating with experts to use more sensitive two-photon excitable probes for analyses of oxygen in vessels in the deeper layers of the healing tissue. We will provide updates to the community as our work progresses.